# Dual Higgs modes entangled into a soliton lattice in CuTe

SeongJin Kwon [1,2], Hyunjin Jung[1,2], SangJin Lee[1,2], Gil Young Cho [1,2], KiJeong Kong[1], ChoongJae Won [1,3,4], Sang-Wook Cheong [3,4,5] & Han Woong Yeom [1,2] ✉

Recently discovered Higgs particle is a key element in the standard model of elementary particles and its analogue in materials, massive Higgs mode, has elucidated intriguing collective phenomena in a wide range of materials with spontaneous symmetry breaking such as antiferromagnets, cold atoms, superconductors, superfluids, and charge density waves (CDW). As a straightforward extension beyond the standard model, multiple Higgs particles have been considered theoretically but not yet for Higgs modes. Here, we report the real-space observations, which suggest two Higgs modes coupled together with a soliton lattice in a solid. Our scanning tunneling microscopy reveals the 1D CDW state of an anisotropic transition metal monochalcogenide crystal CuTe is composed of two distinct but degenerate CDW structures by the layer inversion symmetry broken. More importantly, the amplitudes of each CDW structure oscillate in an out-of-phase fashion to result in a regular array of alternating domains with repeating phase-shift domain walls. This unusual finding is explained by the extra degeneracy in CDWs within the standard Landau theory of the free energy. The multiple and entangled Higgs modes demonstrate how novel collective modes can emerge in systems with distinct symmetries broken simultaneously.

Higgs particle[1] was predicted in 1964 to provide a mechanism for the mass of a few important elementary particles but had been a crucial missing block in the standard model of particle physics until its observation in 2012. It is a bosonic quantum excitation from the Higgs scalar field, which has a Mexican-hat shape potential to break weak isospin symmetry of electroweak interaction. Collective bosonic excitations in a Mexican-hat potential can also arise in solid materials with spontaneously broken symmetry, which are called massive Higgs modes[2]. Studies on massive Higgs modes have elucidated intriguing collective phenomena in a wide range of materials with spontaneous symmetry breaking from ultracold atoms[3–5] to superconductors[6] and charge density wave (CDW) crystals[7]. For example, the observation of

Higgs modes has revealed the existence of unprecedented CDW[8], unusual pairing in cuprate superconductors[9], and nonequilibrium properties of superconductors[10]. While most of those studies have focused on the identification of a single Higgs mode in each material system, the study of the interactions between Higgs modes and other collective modes is only in its infant stage[11–14], and the existence of multiple Higgs modes and their interactions have never been accessed[15,16]. In contrast, the Higgs-to-Higgs interaction is an important test ground for the accuracy of the standard model and the existence of additional Higgs or Higgs-like boson is believed as a gateway to go beyond the standard model. This apparently suggests that the search for such higher-order Higgs interactions can open a route to

[1]Center for Artificial Low Dimensional Electronic Systems, Institute for Basic Science, Pohang 37673, Korea. [2]Department of Physics, Pohang University of Science and Technology, Pohang 37673, Korea. [3]Laboratory for Pohang Emergent Materials, POSTECH, Pohang 37673, Korea. [4]MPPC-CPM, Max Planck POSTECH/Korea Research Initiative, Pohang 37673, Korea. [5]Rutgers Center for Emergent Materials and Department of Physics and Astronomy, Piscataway, NJ 08854, USA. ✉e-mail: yeom@postech.ac.kr

unprecedented emerging phenomena also in solid systems such as the possible coupling of CDW and superconductor Higgs modes for unconventional superconductivity[17].

In the present work, we report on the direct microscopic observation of two Higgs modes, the amplitude oscillations, coexisting to entangle mutually and further with a soliton lattice (phase mode) in a quasi-one-dimensional CDW crystal of CuTe, where the translational symmetry of the crystal is broken one-dimensionally and spontaneously at low temperature. Coexisting Higgs modes are rarely noticed in a solid system[18] while the coupling of a Higgs mode and a phase mode has been predicted and indicated for CDW systems[19] and superconductors[20]. The present theoretical works within the Landau mean-field framework reveal the complementary and oscillatory entanglements of two Higgs modes is a natural consequence of the doublet degeneracy in the CDW structures.

## Results and discussion

### Two degenerate CDW structures

CuTe is a van der Waals layered material and each layer is composed of a wiggled Cu layer sandwiched between two Te layers (Fig. 1a). It has an orthorhombic crystal structure (the $P_{mmn}$ space group symmetry) and lattice constants of $a = 3.149$ Å, $b = 4.086$ Å, and $c = 6.946$ Å[21,22]. This crystal was recently found to undergo a symmetry-breaking phase transition below $T_c = 335$ K[23]. The low-temperature phase has an anisotropically distorted structure with a supercell of $5 \times 1 \times 2$ and the photoelectron spectroscopy study identified a band-gap opening, which clearly indicates the CDW formation[23] (See Supplementary Fig. 1 for our own measurements.). While the driving mechanism of CDW is not fully clear yet[23,24], more recent works on this crystal identified pressure-induced superconductivity[25], which is shared by many CDW materials[26,27], and, to our own particular interest, an unusually stable Higgs mode, that is, an oscillatory mode of the CDW amplitude[28,29].

We characterize this CDW structure and its amplitude fluctuations in atomic scale. The topographic imaging of a CuTe crystal with STM at 78.3 K, where the CDW gap completely opens, finds a well-ordered chain structure (along b-axis) as shown in Fig. 1b. These corrugations indicate that Te atomic rows of the topmost layer are transversely (along a-axis) modulated with a period of $5a_0$ due to the formation of CDW. However, a closer look into the STM image immediately tells that there exist two different corrugation patterns (Fig. 1b). In one pattern, one finds only one bright (higher height or larger local density of states) row in a $5 \times 1$ CDW unit cell (Fig. 1e) but, in the other, two bright rows (Fig. 1f). Two different CDW structures are called $\alpha$ and $\beta$ hereafter and their overall areal ratio is close to 1:1 (Fig. 2). We optimize atomic structures of CuTe in a $5 \times 1 \times 2$ unit cell through extensive DFT total energy calculations and find two distinct atomic structures shown in Fig. 1c as the most stable but energetically degenerated. In the $\alpha$ structure, the top Te rows are coupled transversely in a dimer-dimer-monomer fashion, which in turn forces the bottom Te rows beneath the monomer row to dimerize with the other three in a trimer. This bottom Te layer structure corresponds to the top layer in the other optimized structure of $\beta$. The detailed displacements of Te atoms indicate that $\alpha$ and $\beta$ CDW in fact consist of the same vertical and horizontal components, which are coupled with different relative phase shifts (Fig. 1d). That is, the $\alpha$ and $\beta$ structures are just the opposite faces (Te layers) of the same CDW distortions in the Cu layer, which indicates straightforwardly their energetic degeneracy. The simulated STM images of the DFT atomic structures also reproduce excellently the experimental images as seen in Fig. 1d. The bright protrusions in these empty state STM images (Fig. 1e, f) mainly correspond to the charge-depleted regions between dimers or between dimers and trimers. The charge-accumulated parts, the dimer and monomer chains, instead, appear bright in the filled state images.

### Amplitude and phase fluctuations of degenerate CDWs

Degenerate CDW structures appear as coexisting domains as naturally expected but in an unexpected fashion. The bright and dark stripe domains of a width of about 20 nm appear alternatively and extend longitudinally (Fig. 2a), which correspond to the domains of $\alpha$ and $\beta$ CDW structures, respectively (Fig. 2c). What is very important is that

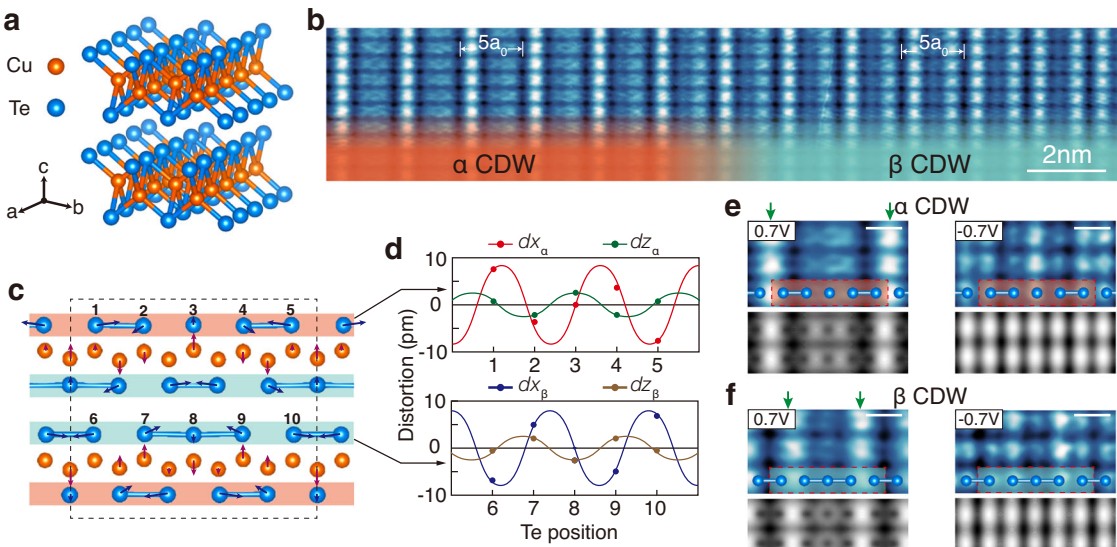

**Fig. 1 | CDW structures in CuTe. a** Atomic structure of a quasi-1D vdW material CuTe. **b** STM topography of well-ordered chains at temperature 78.3 K and sample bias voltage 700 mV. Two distinct STM topographies on the surface Te layer, $\alpha$ (left, red) and $\beta$ (right, green), appear as laterally separated domains. **c** Side view of the atomic structure of a $5 \times 1 \times 2$ CDW unitcell (a dashed box). Different structures emerge alternatively in the top and bottom Te layers sandwiching a distorted Cu layer; the $\alpha$ and $\beta$ structures in the red and blue layer, respectively. **d** Distortions along x and z axes (dx and dz, respectively) of Te atoms in the $\alpha$ and $\beta$ structures. The distortions in $\alpha$ and $\beta$ structures have the same wavevector $q_{CDW}$ and amplitude but opposite directions along a given axis. **e, f** Enlarged STM topographies of $\alpha$ and $\beta$ structures, respectively, and the corresponding DFT simulations at filled ($-700$ mV bias) and empty (700 mV) states. Te atomic structures are overlaid for comparison. The bright lines of protrusions in the empty states (as shown in **b**) are marked with green arrows. The white scale bars represent 0.5 nm in **b** and 0.7 V topographies in **e** and **f** have a 43.8 pm color scale (dark to bright, respectively) while $-0.7$V images have a 20.9 pm color scale.

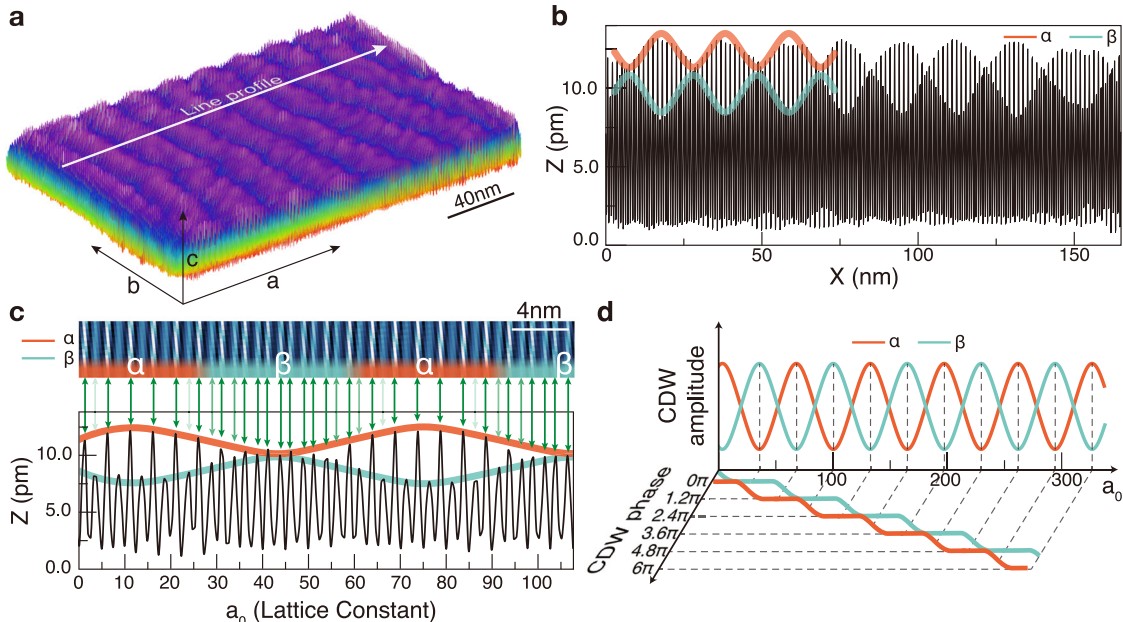

**Fig. 2 | Amplitude and phase modes in the CDW state of CuTe. a** 3D plot of a large scale STM topography at 78.3 K and sample bias voltage 700 mV. The stripy structure represents the alternation of the $\alpha$ (purple) and $\beta$ (blue) domains. **b** Line profile extracted perpendicular to the stripe domains in **a**, whose details are shown in **c**. The CDW maxima of the $\alpha$ structure oscillate in a oscillatory wave and periodically change into the $\beta$ structure whenever the neighboring (second) maxima become comparable in their height. Thus, the amplitude oscillations of the $\alpha$ and $\beta$ CDW can be extracted by the envelope of the first and the second CDW maxima of a CDW unitcell as shown in **b** and **c** with red and blue lines, respectively. **d** $\alpha$ and $\beta$ CDW amplitude oscillations are plotted together with the lateral phase shifts of each CDW, which are extracted as shown in Fig. 3. The entanglement of the amplitude and phase oscillations follows well the prediction of a soliton lattice model (or the phase-amplitude lock-in transition) except for the existence of two amplitude modes.

the CDW amplitude continuously varies to switch between $\alpha$ and $\beta$ phases in a oscillatory pattern (Fig. 2b and c) with the total CDW order (the amount of distortion from the normal phase structure) stays constant (See Fig. 3b and Supplementary Fig. 10). Two CDW fluctuations are fitted excellently with Jacobi elliptic sine functions, which is the general solution of one-dimensional soliton lattices[30,31]. The average wavelength of the amplitude wave is $\lambda_{Higgs} = 24$ nm with the standard deviation 2.9 nm, which is sufficiently longer than the CDW wave length $\lambda_{CDW} = 1.6$ nm. The amplitude variation reaches roughly half of the maximal CDW amplitude (Fig. 3a). Moreover, if one measures the difference of the phase of CDW between neighboring $\alpha$ (or $\beta$) domains, one can get a constant phase shift of $3a_0$ or $6\pi/5$ over the whole surface area measured (Fig. 3c). We thus can conclude that the two different CDW structures have their own amplitude oscillations, in other words, Higgs modes, which have the same wavelength but opposite phase. At the same time, these Higgs modes are coupled with phase-shift domain walls and are frozen into the CDW domain landscape indicating the formation of a soliton lattice. Similar degenerate CDW structures were observed in well-known quasi 2D CDW crystals of 2H-NbSe$_2$ and 2H-TaSe$_2$[32–35], which coexist as inhomogeneous domains with phase domain walls. These domains are however not regular and do not exhibit a continuous CDW amplitude variation. On the other hand, the nearly commensurate CDW phase of 1T-TaS$_2$ with a regular 2D domain wall network was previously attributed to an amplitude mode[36,37]. However, the domain wall lattice of two distinct CDWs or the two CDW amplitude modes is unique. The present system is thus unconventional in a sense that the CDW ground state is composed of Higgs and phase modes.

## Origin of the domain formation with two Higgs and one phase modes entangled

The origin of the emergence of unusual Higgs and phase modes can be explained by the phenomenological theory. We use a well-established Landau-Ginzburg model with two period-5 CDW order parameters,

$\{\phi_\alpha(x), \phi_\beta(x)\}$. Under the lattice translation, they have a 5-fold degeneracy as $\mathbb{Z}_5 : \phi_j \to \phi_j e^{2\pi i/5}$. At the same time, they are mapped to each other by the layer-inversion symmetry (more precisely, the mirror glide symmetry) $\mathbb{Z}_2 : \phi_\alpha \leftrightarrow \phi_\beta$, which corresponds to $\alpha$ and $\beta$ CDW structures. Hence, our theory has the $\mathbb{Z}_5 \times \mathbb{Z}_2$ symmetry, whose free energy density is given by

$$f = \sum_{i=\alpha,\beta} \left( \frac{K}{2} \left| \frac{\partial \phi_i}{\partial x} \right|^2 + \frac{a}{2} |\phi_i|^2 + \frac{b}{5} (\phi_i^5 + \phi_i^{*5}) + \frac{d}{6} |\phi_i|^6 \right) + g_2 \left( \phi_\alpha^* \phi_\beta + c.c \right) + g_4 |\phi_\alpha|^2 |\phi_\beta|^2. \tag{1}$$

Here, $c.c.$ is the complex conjugate and the coefficients $\{K, a, b, d \cdots\}$ are in general functions of external parameters, whose detailed values are irrelevant to our main conclusion. We will set $K = 1$ for convenience and $d > 0$ to make the free energy bounded below the CDW transition temperature, $a(T) \propto (T - T_{CDW}) < 0$[23]. Since the experiment indicates the spontaneous breaking of the $\mathbb{Z}_2$ symmetry in the ground states, i.e., it is either in the $\alpha$ or $\beta$ CDW state (Fig. 4a), $g_4$ should be large and positive so that two CDWs repel each other. Furthermore, the sum of the amplitudes is experimentally found to be almost constant in space (See Fig. 3b, and Supplementary Fig. 10) and hence we impose $|\phi_\alpha(x)| + |\phi_\beta(x)| = A_0 > 0$. This simply means that the total charge per unit cell should be shared between $\alpha$ and $\beta$ states. In order to generate oscillating solutions within the finite-size simulation, we set boundary conditions; the system is exclusively in the $\alpha$ ($\beta$) CDW state at $x = 0$ (at the other end), that is, $|\phi_\alpha| = A_0, |\phi_\beta| = 0$ ($|\phi_\alpha| = 0, |\phi_\beta| = A_0$) corresponding to the red (green) circle in the free energy landscape in Fig. 4a, b and c. Then, the equations of motion derived from the free energy Eq. (1) is solved numerically (See Supplementary Information for a detailed description) to yield the oscillations of both amplitudes and phases in consistency with the experimental data (Fig. 4e and f). In particular, two CDW orders crossover each other smoothly, and their amplitudes oscillate with the

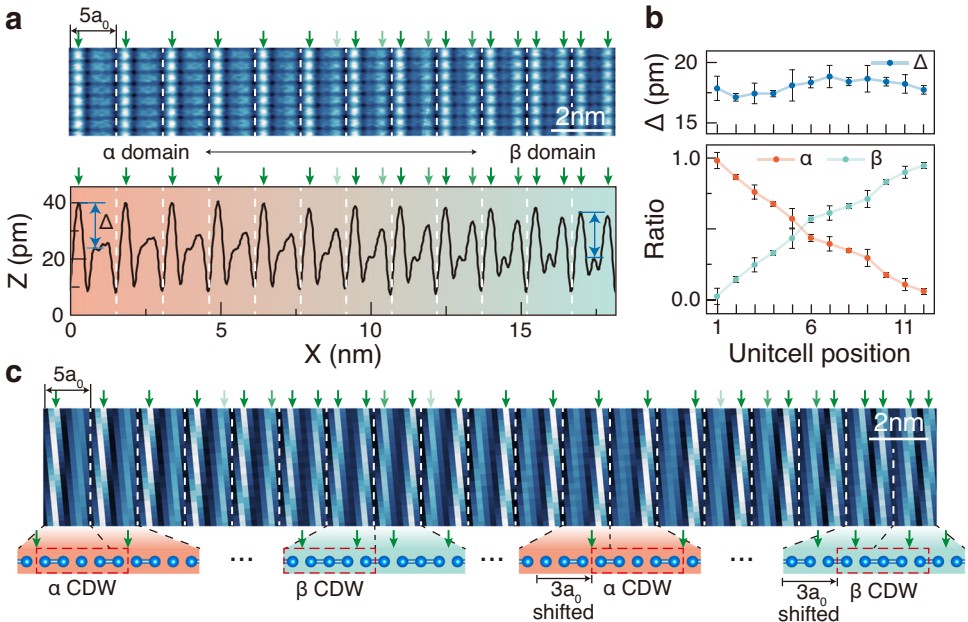

**Fig. 3 | CDW phase shift accompanying the dual amplitude modes. a** $\alpha$ (red)-$\beta$ (green) domain boundary topography and its line profile at bias voltage +500 mV. **b** CDW order parameter, as defined by the difference of the height difference of the high and low height chains in the line profile, and the ratio of $\alpha$ and $\beta$ structures are plotted across one domain boundary. Average order parameter and the ratio are obtained with their error bars along the seven $\alpha$-$\beta$ lines shown in **a**. **c** STM topography crossing a few $\alpha$ and $\beta$ domains. The neighboring $\alpha$ ($\beta$) CDW structure is $3a_0$ translated, which is due to the different atomic pairing of the $\alpha$ and $\beta$ CDW structures, 2-1-2 and 3-2 respectively. The height differs by 14.05pm, from the dark to bright, respectively.

$\pi$ phase shift, and domain walls occur when the dominant CDW orders are changed from $\alpha \rightarrow \beta$ or $\beta \rightarrow \alpha$.

Heuristically, the oscillations of amplitudes can be understood from the free energy landscape (dashed lines in Fig. 4a, b), which is similar to the well-known behaviors of the real scalar $\phi^4$ theory[30,31,38]. The energy landscape of this theory is generated by the potential $V(\phi) \propto (\phi^2 - \langle \phi \rangle^2)^2$ whose ground states are $\phi = \pm \langle \phi \rangle$ and the excited state is known as the soliton lattice solution. This solution essentially describes a particle moving along the potential $-V(\phi)$, which starts from the point slightly below one hill (one ground state) and ends at the other counterpart (the other ground state) (Fig. 4c). The time evolution of the particle corresponds to the spatial evolution of $\phi(x)$ (See Fig. 4d). In our case, the amplitude oscillations in space are generated by the system's trajectory between the two ground states labeled by $(\langle \phi_\alpha \rangle \neq 0, \langle \phi_\beta \rangle = 0)$ and $(\langle \phi_\alpha \rangle = 0, \langle \phi_\beta \rangle \neq 0)$. When the amplitude of the CDW is relaxed, the phase of the CDW can fluctuate easily and thus experience the domain walls, where the corresponding order parameter almost vanishes, to yield the intertwined phase and amplitude oscillations (Fig. 4e, f). This phenomenon has been predicted for a single CDW amplitude mode[19] but its material realization has been elusive. The present case corresponds to its non-trivial extension to two competing CDWs since the two amplitude modes are entangled with one phase mode. As noted above, this unconventional CDW originates from the extra symmetry breaking to lead to the $\mathbb{Z}_5 \times \mathbb{Z}_2$ symmetry.

While our observation is limited to the spatial variations of amplitude and phase modes, their connection to dynamical modes is not excluded. If one includes the time-derivative terms in theory, they naturally exhibit temporal oscillations[31,39]. We can further note that the amplitude modes and the soliton lattice seem to be laterally pinned by atomic scale defects in the present system (See Supplementary Fig. 11). On the other hand, the recent optical measurements observed the amplitude mode as excited by laser pulses[28,29]. The dynamics of the present amplitude and phase variations are however not directly observed in the present work and invite further study at an elevated temperature.

Our study presents the STM experiments to map the amplitude and phase modulations of quasi-one-dimensional CDWs in CuTe. Dual Higgs (amplitude) modes entangled mutually and with the soliton lattice (phase mode) are discovered and its microscopic description within the standard mean-field theory is obtained. While the competing coexistence of superconducting and CDW amplitude modes was observed previously[18], the entangled dual Higgs modes and the direct observation of the Higgs and phase modes entangled have not been reported. The extra degeneracy of the energetically equivalent CDW structures plays a key role in the emergence of entangled collective modes. We further expect that the presented mechanism can be extended to other systems with multiple competing ground states to discover novel composites of collective modes. In particular, the interactions between Higgs and phases modes can unravel spatial variations of domains and domain walls, which are ubiquitous in various CDW phases of transition metal chalcogenides.

## Methods

### Crystal growth
Single crystals of CuTe were grown by typical self-flux methods under Te-rich compositions. Copper powder (99.999%, Alfa Aesar) and Te pieces (99.999%, Alfa Aesar) were put in the alumina crucible with a frit disc, then sealed in Ar-gas purged evacuated quartz tube. The quartz ampule was heated at 600 °C for 12 h, then slowly cooled down to 450–350 °C for 1–2 °C per hour. Then, the ampule was centrifuged to separate the single crystal from the flux. The single crystal structure and purity was checked by x-ray diffraction (See Supplementary Fig. 13).

### STM measurements
A commercial cryogenic STM (Unisoku, Japan) is used for STM and STS experiments. CuTe crystal is cleaved at the pressure below $5 \times 10^{-10}$ torr to obtain clean surface layers. All measurements were performed at 78.3 K using platinum-iridium tips cut mechanically. The STM topographies are modified with a 2D-FFT and mean-value correction to reduce a high-frequency noise and tilting of the crystal.

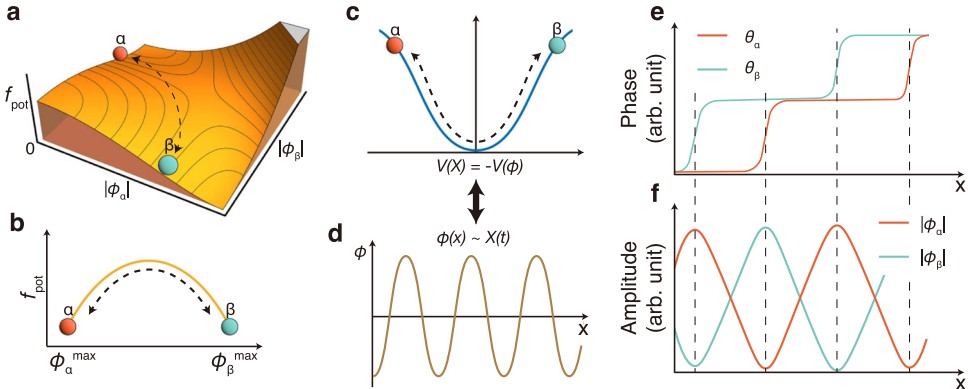

**Fig. 4 | Phenomenological model of the amplitude and phase oscillations.**
**a** Free energy landscape of two degenerate CDW states. **b** Free energy change of oscillating CDWs along $|\phi_\alpha| + |\phi_\beta| = A_0$. **c, d** The spatial evolution of the CDW order parameters under $V(\phi)$ can be interpreted as the time evolution of a particle subject to the potential $-V(X)$[30]. **e, f** Results of numerical simulations for two CDW states. Top and bottom panels correspond to the evolution of the phases and the amplitudes of each state, respectively.

## Computations

DFT calculations were carried out using the Vienna ab-initio simulation package, at the level of generalized-gradient approximation (GGA)[40,41]. We used the projector augmented wave method for the description of the core-valence interaction. The van der Waals (vdW) interaction has been taken into account by DFT-D3 method with Becke-Jonson damping[42]. The cutoff energy was set as 400 eV. For structural relaxations, the k-point samplings of Brillouin zones used Γ-centered k meshes of size $20 \times 16 \times 8$ and $4 \times 16 \times 4$ for the non-CDW and the CDW structure, respectively. The energy and force convergence criteria were set to be $10^{-6}$ eV and 0.01 eV/Å, respectively.

We also performed the GGA + U calculations to account for the electron correlation in Cu $d$ orbitals with the Dudarev method[43]. The modulated CDW structure becomes stable when the Coulomb correlation for Cu $d$ orbitals is considered. The hybridization between Te and Cu states is reduced when the on-site Coulomb interaction for the Cu $d$ states are included, which is consistent with the previous DFT result of Kim et al.[24]. We have adopted $U_{eff} = U - J = 9$ eV, which gives 39 meV energy gain for the CDW phase and 0.2 Å asymmetry in interatomic distance between nearest neighbor Te atoms ($d_{max} = 3.18$ Å, $d_{min} = 2.99$ Å).

## Angle-resolved photoelectron spectroscopy measurements

We performed the temperature-dependent ARPES measurements at Beamline 4A2 of the Pohang Light Source with a DA-30-L analyzer (Omicron-Scienta). Samples were cleaved in an ultrahigh vacuum chamber at a temperature of 75 K and then heated to temperatures of 205, 310, 330, and 350 K. We used linearly polarized undulator photon beam of an energy of 80 eV in Supplementary Fig. 1a and 100 eV in Supplementary Fig. 1b–g.

## Data availability

The source data used for Figs. 1, 2, 3, and 4 are fully available on request from the authors and are provided as in Source data file with this paper [https://doi.org/10.6084/m9.figshare.24848139]. Source data are provided with this paper.

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

## Acknowledgements

This work was supported by the Institute for Basic Science (Grant No. IBS-R014-D1). C.J.W. was supported by the National Research Foundation of Korea funded by the Ministry of Science and ICT (Grant No. 2022M3H4A1A04074153). S.W.C. was funded in part by the Gordon and Betty Moore Foundation's EPiQS Initiative through Grant GBMF4413 to the Rutgers Center for Emergent Materials.

## Author contributions

S.J.K. performed S.T.M. experiments. H.J.J. performed ARPES experiments. S.J.L. and G.Y.G. performed theoretical simulations. K.J.K. performed DFT calculations. C.J.W. synthesized the crystal under the supervision of SWC. H.W.Y. conceived the main ideas and supervised the whole project. S.J.K., G.Y.C., and H.W.Y. wrote the manuscript.

## Competing interests

The authors declare no competing interests.
