## [Peer Review File · Nature Communications]

REVIEWER COMMENTS

Reviewer #1 (Remarks to the Author):

Review report for NCOMMS-23-28966-T, manuscript titled, "Dual Higgs modes entangled into a soliton lattice observed in CuTe", by Kwon et al, submitted for review purpose in Nature Communications.

The manuscript has discussion about the experimental observation of two Higgs modes coupled with a soliton lattice and associated with charge density wave (CDW) in CuTe. The entangled Higgs mode and amplitude oscillations in a quasi-one-dimensional CDW crystal of CuTe, are interesting. Authors have discussed the multiple and entangled Higgs mode with two CDW structures (α and β explained by the broken symmetry), which have been observed in scanning tunneling microscopy supported by DFT calculations. Authors have modeled the observations using Landau-Ginzburg model to explain the origin of CDW domain formation in CuTe.

The results are important in understanding the competing superconductivity and charge density wave phenomena in layered chalcogenide materials. The article is well written with all the details of experimentations. However, following are some questions and suggestions, which can be addressed and clarified before the publication.

1. Generally, the CDW is understood in the realm of Peierls distortion, Fermi surface nesting, [npj 2D Materials and Applications 7, 35 (2023)] electron-phonon coupling [Phys. Rev. Res., 02, 033118 (2020)]. Though the Higgs modes are discussed in detail yet some more discussion is required on how these modes can unravel the CDW mechanism for other chalcogenides?

2. Author have depicted the two different CDW structures, α and β CDW in CuTe, which are coupled with relative phase shifts. However, it is not clear if the different CDW structures can coexist at different temperatures as well? Which one is more stable (either α or β) and why?

3. Author have discussed the ARPES spectra of CuTe in a broad temperature range which showed the opening of band gap. Is it possible to tell the exact CDW and superconducting transition temperature for CuTe? The major reason, behind the curiosity, is selection of ~ 78.3 K for capturing the STM topographic images.

4. Samples are prepared by self-flux method and CuTe has an interesting phase diagram (ACS Appl. Energy Mater. 2020, 3, 3, 2175) with various structural possibilities, which makes the preparation of pure CuTe as a very challenging task. As the global structural characterization are not given, I am curious if there are any traces of Cu₂Te or other phases? There are several reports on observation of Higgs modes during CDW formation using Raman spectroscopy and THz pump-probe spectroscopy. Raman spectroscopy can be a good tool to support with STM to confirm the origin of amplitude modes as Higgs modes. Authors may like to comment/discuss on the possibilities if measurements are not possible.

5. Based on the modulated periodicity, the CDW can be classified to incommensurate and commensurate. Thus, there are some curious queries, (i) Do the Higgs modes have any correlation with (in-) commensurate CDW phases? (ii) For CuTe, what is the nature of the CDW and is it possible to determine CDW ordering vector for α or β phases? Are Higgs mode associated with phase mode or amplitude mode in CuTe and why?

6. Minor comment: In Figure 4, sequencing (a, b, c..) can follow other figures in the manuscript.

Reviewer #2 (Remarks to the Author):

The manuscript by Kwon et al. reports scanning tunneling microscopy study of a prototypical 1D CDW material CuTe. Interestingly, they observed two degenerate CDW structures, and the continuous spatial variation of the CDW amplitudes which lead to a regular array of alternating domains. The authors attributed these observations as results of entangled Higgs modes. The experimental observations are interesting and may bring new insights to the understanding of CDWs and other collective phenomena. However, there are some issues, which should be properly addressed before consideration for publication in Nature Communications.

1. The major concern is about the scientific interpretation of “Higgs mode” for the experimental results. Higgs mode (or amplitude mode for CDW systems) is the collective excitation of the CDW condensate. It usually appears as periodic oscillation over time in time-domain measurements such as pump-probe spectroscopy, or it can be a peak at finite frequency in frequency-domain measurements such as Raman spectroscopy. In this work, the authors observed the static spatial variation of the CDW amplitude, which is like a standing wave in space formed owing to the CDW ground state degeneracy. This phenomenon is novel, but I doubt the reasonability of claiming this as manifestation of dynamical “Higgs modes”, which are collective excitations and should oscillation in time domain.

Besides, the authors used a phenomenological GL model to simulate their results. However, the model shown in the manuscript only describes the static situation. The model with two entangled CDW order parameters gives a state with spatially varying CDW amplitudes, but it has no connection to the Higgs modes.

In all, even though there exist two CDW structures, with spatial amplitude oscillations forming alternating domains, this work do not provide “direct microscopic observation of two Higgs modes” as written in the 2nd paragraph in page 3. If the authors consist on this interpretation of “Higgs modes”, they should explain much more explicitly and clearly why and how the Higgs modes are manifested here.

2. Disorders usually pose strong pinning effects on CDWs, which have become a research focus in recent years [e.g., Liu et al. PRL 126, 256401 (2021); Yue et al., Nature Communications 11, 98 (2020); Oh et al., PRL 125, 036804 (2020)]. The authors should discuss about how the disorders affects the CDW patterns in the main text (instead of just simply discussing in the supplementary material). This may be of interest of many readers.

3. In the 2nd paragraph on page 3, the authors write: “The coexisting Higgs modes are observed for the first time in a solid system to the best of our knowledge”. To my knowledge, coexisting Higgs modes were reported before. For example, the paper Phys. Rev. B 89, 060503(R) shows coexisting CDW amplitude (Higgs) mode and superconducting Higgs mode.

4. There are some typographical errors in the manuscript. In the first paragraph, the reference index jumps from 2 to 5, and from 8 to 11, etc. The authors should carefully check the typography.

Reviewer #3 (Remarks to the Author):

Reviewer #4 (Remarks to the Author):

The layered material CuTe is known for its $5 \times 1 \times 2$ charge density wave (CDW) instability leading to quasi-1D chain-like substructures within the 3D crystal. Looking at the surface of the crystal at low temperatures with a scanning-tunneling microscopy (STM), one can directly visualize the CDW.

Furthermore, two different surface patterns (called alpha and beta CDW structure in the current manuscript) can be observed, corresponding to the two facets of a bilayer sub-slab in the CDW. The two CDW phases are degenerate.

The new finding presented in the current manuscript is a periodic long wavelength (24 nm) variation of the CDW amplitude and, interestingly, the dual presence of both CDW patterns in an alternating fashion. The finding is described with a phenomenological Landau-Ginzburg type model respecting the symmetry of the problem and using two complex order parameters for the two CDW phases. An analogy to the presence of two entangled Higgs modes in high-energy physics is drawn.

The experimental data is presented in a convincing way. The findings are new and quite spectacular (even without reference to "Higgs physics"). The only criticism I have is that the phenomenological model only describes but does not explain the occurrence of the "dual Higgs modes". Can the occurrence of this phenomenon be derived from the electronic and vibrational structure of CuTe? Or, in other words, can the parameters in the phenomenological model be calculated starting from the atomic structure of the crystal, i.e., from first principles?

If the authors could provide a more profound explanation of their findings, that would greatly enhance the impact of this manuscript. But in any case, I recommend publication of the findings in Nature Communications.

Reply to the Comments of Reviewer #1

Comment 1) The manuscript has discussion about the experimental observation of two Higgs modes coupled with a soliton lattice and associated with charge density wave (CDW) in CuTe. The entangled Higgs mode and amplitude oscillations in a quasi-one-dimensional CDW crystal of CuTe, are interesting. Authors have discussed the multiple and entangled Higgs mode with two CDW structures (α and β explained by the broken symmetry), which have been observed in scanning tunneling microscopy supported by DFT calculations. Authors have modeled the observations using Landau-Ginzburg model to explain the origin of CDW domain formation in CuTe.

The results are important in understanding the competing superconductivity and charge density wave phenomena in layered chalcogenide materials. The article is well written with all the details of experimentations. However, following are some questions and suggestions, which can be addressed and clarified before the publication.

Generally, the CDW is understood in the realm of Peierls distortion, Fermi surface nesting, [npj 2D Materials and Applications 7, 35 (2023)] electron-phonon coupling [Phys. Rev. Res., 02, 033118 (2020)]. Though the Higgs modes are discussed in detail yet some more discussion is required on how these modes can unravel the CDW mechanism for other chalcogenides?

(Our reply) We appreciate this comment. Generally speaking, it has been rather well understood that the Peierls mechanism based on Fermi surface nesting cannot be a major driving force for most CDW systems but the electron-phonon interaction, broadly speaking, aided by other types of interaction has to be considered. Up to now, there are a few publications on the mechanism of the CDW of CuTe, which discusses the electron-phonon coupling aided by the Fermi surface nesting and the Coulomb interaction [Refs. 29 and 30 in the main text] but there is no clear-cut conclusion yet. However, it has to be noted that the Higgs or the amplitude mode is a well-established excitation from the CDW ground state irrespective of the driving force of CDW itself. The CDW driving mechanism is on the energy scale of the CDW energy gap while the excitations from the CDW state must have a much lower energy scale, naturally. Note also that the current description of the amplitude mode is based on the Landau-Ginzburg mean field theory, where the

microscopic interaction is irrelevant. We briefly mentioned the current discussion of the CDW mechanism of CuTe in the revised manuscript with the updated references.

Comment 2) Author have depicted the two different CDW structures, α and β CDW in CuTe, which are coupled with relative phase shifts. However, it is not clear if the different CDW structures can coexist at different temperatures as well? Which one is more stable (either α or β) and why?

(Our reply) We appreciate this important comment. We have checked the stability (the total energy) of both phases with DFT calculations and they are fully degenerate in their energies as already mentioned in the manuscript. This is very important for the formation of the present dual Higgs modes, which is based on the degeneracy of the two structures as stated in the manuscript. We have also measured the total areal ratio of the two structures, as mentioned in the manuscript, which is close to 1:1 within the experimental accuracy confirming their energetic degeneracy. We thus can naturally expect that these two structures would coexist for the whole temperature range. Note also that the coexistence of two different CDW structures are well established in the prototypical CDW systems of, for example, NbSe₂ [Ref. 39 in the main text].

Comment 3) Author have discussed the ARPES spectra of CuTe in a broad temperature range which showed the opening of band gap. Is it possible to tell the exact CDW and superconducting transition temperature for CuTe? The major reason, behind the curiosity, is selection of ~ 78.3 K for capturing the STM topographic images.

(Our reply) We appreciate this comment. The present material does not go into a superconducting phase in its pristine state [Ref. 34 in the main text] and the CDW transition temperature is as high as 335 K with a very gradual gap opening [Ref. 29 in the main text]. The transition saturates at around 80 K, which is the reason for us to choose 78 K. We briefly mention this in the revised manuscript. (This is also the easiest choice when one use LN2 cooling.)

Comment 4) Samples are prepared by self-flux method and CuTe has an interesting phase diagram (ACS Appl. Energy Mater. 2020, 3, 3, 2175) with various structural possibilities, which makes the preparation of pure CuTe as a very challenging task. As the global structural characterization are not given, I am curious if there are any traces of Cu₂Te or other phases?

(Our reply) We appreciate this careful comment but the Cu_2Te phase is not possible in our crystal growth. The above-mentioned reference is a polycrystalline sample made in a totally different method from our own. Our self-flux growth uses Te-rich compositions and there is no Cu_2Te phase in Te-rich side of the Ce-Te phase diagram. Our x-ray data shown below and in the revised supplements clearly indicate a single crystal of a pure CuTe phase. This extra information would be helpful to readers.

Figure caption) XRD (two theta scan) data for the current CuTe crystal. These peaks exactly match with the structural data of CuTe.

There are several reports on observation of Higgs modes during CDW formation using Raman spectroscopy and THz pump-probe spectroscopy. Raman spectroscopy can be a good tool to support with STM to confirm the origin of amplitude modes as Higgs modes. Authors may like to comment/discuss on the possibilities if measurements are not possible.

(Our reply) We appreciate this comment. As we already mentioned in the manuscript, there is a Raman study on the present sample, which found the signature of a Higg mode [Ref. 31 in the main text]. We also noticed a recent work with THz pump-probe spectroscopy [R.S. Li, Phys.

Rev. B 105, 115102 (2022)], which agrees well with the Raman study for the existence of the Higgs mode. We added this reference in the revised manuscript.

Comment 5) Based on the modulated periodicity, the CDW can be classified to incommensurate and commensurate. Thus, there are some curious queries, (i) Do the Higgs modes have any correlation with (in-) commensurate CDW phases? (ii) For CuTe, what is the nature of the CDW and is it possible to determine CDW ordering vector for α or β phases? Are Higgs mode associated with phase mode or amplitude mode in CuTe and why?

(Our reply) We appreciate this insightful comment. As the reviewer suggested, the present system is an incommensurate CDW phase due to the presence of the regular phase-shift domain walls (or soliton lattice). The domain wall periodicity is the same as the periodicity of the amplitude mode since the domain wall (soliton) lattice and the amplitude mode are perfectly entangled. Note also that a Higgs mode is by definition an amplitude model and the soliton lattice by definition a phase mode. The entanglement of a phase mode and an amplitude mode is well established in theory as mentioned in the manuscript [Ref. 25 in the main text]. The entanglement of the different CDW phases (thus their amplitude) with phase domain walls was well characterized in our previous work on NbSe₂ [Ref. 39 in the main text].

Comment 6) Minor comment: In Figure 4, sequencing (a, b, c..) can follow other figures in the manuscript.

(Our reply) We are grateful for this helpful suggestion. The figure was accordingly modified.

Reply to the Comments of Reviewer #2

Comment 1) The manuscript by Kwon et al. reports scanning tunneling microscopy study of a prototypical 1D CDW material CuTe. Interestingly, they observed two degenerate CDW structures, and the continuous spatial variation of the CDW amplitudes which lead to a regular array of alternating domains. The authors attributed these observations as results of entangled Higgs modes. The experimental observations are interesting and may bring new insights to the understanding of CDWs and other collective phenomena. However, there are some issues, which

should be properly addressed before consideration for publication in Nature Communications.

1. The major concern is about the scientific interpretation of “Higgs mode” for the experimental results. Higgs mode (or amplitude mode for CDW systems) is the collective excitation of the CDW condensate. It usually appears as periodic oscillation over time in time-domain measurements such as pump-probe spectroscopy, or it can be a peak at finite frequency in frequency-domain measurements such as Raman spectroscopy. In this work, the authors observed the static spatial variation of the CDW amplitude, which is like a standing wave in space formed owing to the CDW ground state degeneracy. This phenomenon is novel, but I doubt the reasonability of claiming this as manifestation of dynamical “Higgs modes”, which are collective excitations and should oscillate in time domain.

Besides, the authors used a phenomenological GL model to simulate their results. However, the model shown in the manuscript only describes the static situation. The model with two entangled CDW order parameters gives a state with spatially varying CDW amplitudes, but it has no connection to the Higgs modes.

In all, even though there exist two CDW structures, with spatial amplitude oscillations forming alternating domains, this work does not provide “direct microscopic observation of two Higgs modes” as written in the 2nd paragraph in page 3. If the authors insist on this interpretation of “Higgs modes”, they should explain much more explicitly and clearly why and how the Higgs modes are manifested here.

Our reply) We appreciate this important comment. However, we do not agree with the reviewer’s opinion on the Higgs (amplitude) mode. Amplitude and phase modes of CDW are based on the spatial variation of the CDW amplitude and phase by definition. As an excited state from the CDW ground state, of course, these modes would have dynamics. The excited state amplitude mode of the present system was already observed by optical measurement as excited by laser [S. Wang, et al., Appl. Phys. Lett. 120, 151902 (2022) and R.S. Li, Phys. Rev. B 105, 115102 (2022)]. At the same time, the amplitude or phase mode can be locked or pinned in space as observed in various systems. For the latter, the soliton lattice is well known. Even for an individual soliton excitation, it has a dynamic and kinetic nature by itself but can be easily pinned in space. One does not distinguish a pinned soliton (or stationary soliton lattice) and a moving soliton (or dynamic phase mode) fundamentally [Kim et al., Nature Physics 13, 444 (2017) and Park et al., Nature Nanotechnology 17, 244 (2022)]. Along the same logic, there should be no fundamental difference between the dynamic amplitude mode and its pinned version. The spatial pinning of excited states is not limited to the CDW systems but is ubiquitous. Note also that the CDW itself

has a dynamic nature and is not necessarily fixed in space but in many cases pinned by defects. It can be depinned by excitation to form the well-known CDW current. We thus think that the amplitude mode observed by the laser excitation is the depinned version of the pinned or frozen mode we observe by STM.

On the other hand, in terms of theory, the only reason for us to consider the static Ginzburg-Landau theory is that we wanted to explain the spatial oscillations in our STM data. However, this does not mean that the corresponding mode is static in time. If one includes the dynamical term in the free energy [1,2]

$$f_{dynamic} = \sum_{i=1,2} \frac{1}{2} \left(\frac{\partial \phi_i}{\partial t} \right)^2 + f(\phi_1, \phi_2)$$

where $f(\phi_1, \phi_2)$ is the free energy density in the previous version of our manuscript. When one performs the variation with respect to the CDW fields, one obtains the standard spatio-temporal differential equations for the order parameters. Such spatio-temporal differential equation can be solved by “separation of variables” and our previous differential equation corresponds to the solution of “spatial part” of the full solution. When it is plugged back to the full time-dependent equation, it gives rise to the simple temporal oscillation. This is a standard trick in solving, for example, the Schrödinger’s equations, and the soliton dynamics [for example, Lizunova, M. & van Wezel, J. SciPost Phys. Lect. Notes 23(2021) and Faddeev, L. D., & V. E. Korepin, Phys. Rep. 42, 3 (1978).] The fundamental issue is thus not the existence of the time variation but how the mode can be the mode pinned. In this respect, the pinning mechanism of an amplitude mode through the entanglement of a phase mode has been theoretically well-established since the early 1980’s as mentioned in our manuscript. The spatially fixed phase modes and amplitude modes are thus in fact quite common for incommensurate CDW phases.

Reflecting the concern of the reviewer and possibly of the readers, we explicitly state that we assume the present Higg mode as a spatially pinned one by defects. The experimental evidence of defect pinning is discussed below and in the revised supplements.

Comment 2) Disorders usually pose strong pinning effects on CDWs, which have become a research focus in recent years [e.g., Liu et al. PRL 126, 256401 (2021); Yue et al., Nature Communications 11, 98 (2020); Oh et al., PRL 125, 036804 (2020)]. The authors should discuss

about how the disorders affects the CDW patterns in the main text (instead of just simply discussing in the supplementary material). This may be of interest of many readers.

Our reply) We appreciate this insightful comment and grateful to the reviewer to connect naturally the issue (the previous comment) to the defect issue. Reflecting the reviewers' opinion, we mention this issue in the main text of the revised manuscript and provide more detailed analysis in the supplements (see also figure below).

As summarized in the figures below (also included in the revised supplements as Fig. S11), there are mainly two different types (A and B) of defects and they have a rather weak pinning effect on the soliton lattice and the amplitude mode.

Figure caption) (a) Large-scale STM image (bias 700 mV) showing the positions of two major types of defects (A and B defects in yellow and red circles, respectively) and the amplitude modes (α and β phases in bright and dark contrast, respectively). (b) Enlarged STM images (bias 500 mV) of A and B defects with their relative populations within a half period of the amplitude wave [from the maximum of the α (β) phase to the neighboring minimum of it]. Type A defects favor being located at the maximum or minimum of α (or β) phase while type B defects have a weaker tendency to be located in the transition region between the two phases. (c) Atomic scale location of the A and B type defects within the α and β structures. The atomic structures of the defects are not decided at present.

Comment 3) In the 2nd paragraph on page 3, the authors write: “The coexisting Higgs modes are observed for the first time in a solid system to the best of our knowledge”. To my knowledge, coexisting Higgs modes were reported before. For example, the paper Phys. Rev. B 89, 060503(R) shows coexisting CDW amplitude (Higgs) mode and superconducting Higgs mode.

Our reply) We deeply appreciate this comment and apologize for our limited knowledge. We mention this reference in the revised manuscript and toned down our claim.

Comment 4) There are some typographical errors in the manuscript. In the first paragraph, the reference index jumps from 2 to 5, and from 8 to 11, etc. The authors should carefully check the typography.

Our reply) We are grateful for this comment and corrected those typos and others throughout the manuscript.

Reply to the Comments of Reviewer #4

Comment 1) The layered material CuTe is known for its $5 \times 1 \times 2$ charge density wave (CDW) instability leading to quasi-1D chain-like substructures within the 3D crystal. Looking at the surface of the crystal at low temperatures with a scanning-tunneling microscopy (STM), one can directly visualize the CDW. Furthermore, two different surface patterns (called alpha and beta CDW structure in the current manuscript) can be observed, corresponding to the two facets of a bilayer sub-slab in the CDW. The two CDW phases are degenerate.

The new finding presented in the current manuscript is a periodic long wavelength (24 nm) variation of the CDW amplitude and, interestingly, the dual presence of both CDW patterns in an alternating fashion. The finding is described with a phenomenological Landau-Ginzburg type model respecting the symmetry of the problem and using two complex order parameters for the two CDW phases. An analogy to the presence of two entangled Higgs modes in high-energy physics is drawn.

The experimental data is presented in a convincing way. The findings are new and quite spectacular (even without reference to "Higgs physics"). The only criticism I have is that the phenomenological model only describes but does not explain the occurrence of the "dual Higgs modes". Can the occurrence of this phenomenon be derived from the electronic and vibrational structure of CuTe? Or, in other words, can the parameters in the phenomenological model be calculated starting from the atomic structure of the crystal, i.e., from first principles?

(Our reply) We appreciate this important comment. Our model calculation is very much generic and depends only on a few symmetry constraints and the degeneracy of the system, which are obvious in the experimental data and DFT calculations. This phenomenon however basically comes from the existence of two different 5×1 CDW structures whose energies are degenerate. In turn, this unique structural and energetic degeneracy should come from the electronic and atomic structures of CuTe.

Comment 2) If the authors could provide a more profound explanation of their findings, that would greatly enhance the impact of this manuscript. But in any case, I recommend publication of the findings in Nature Communications.

(Our reply) We appreciate this comment. We have tried to explain the profound meaning and implication of the present work in the discussion section of the revised manuscript.

REVIEWER COMMENTS

Reviewer #1 (Remarks to the Author):

The authors have clarified the doubts and made good changes to the manuscript, which is more clearer from readers.

The explanation for one of the comment is partially satisfactory, for instance, how Higgs modes can unravel the CDW mechanism for other chalcogenides? The author can relook at the comment.

Rest is fine.

Reviewer #2 (Remarks to the Author):

The experimental results of this manuscript are meaningful and could be of interest of research from CDW and superconductivity communities, but there are still some concerns which I think are not addressed properly in the revised manuscript.

1. I disagree with some of the authors' understanding about Higgs modes. In the first paragraph of Reply to the Comments of Reviewer #2, the authors write: "Amplitude and phase modes of CDW are based on the spatial variation of the CDW amplitude and phase by definition." Is there any literature supportive of such claim? In my understanding, the definition of Higgs modes in CDW or superconducting systems is the collective excitation of the order parameter, which indeed is dynamical variation along time which in principle cannot be characterized by the static STM probe. According to its definition, Higgs modes have no direct relation to the static spatial distribution. If the Higgs mode could be related to the spatial variation of CDW amplitude, it should be in an indirect way, which in this work is due to the pinning of CDW amplitude. Besides, the authors also write: "Along the same logic, there should be no fundamental difference between the dynamic amplitude mode and its pinned version." I think such "logic" is not solid. The two modes are actually different. Phase modes are zero-energy excitations and should be easy for pinning. But Higgs modes have finite energy. In this work, it seems that the degeneracy of two CDW structures are helpful for easier pinning and lead to the intriguing results.

Therefore, if the authors interpret the results in the scenario of Higgs modes, they should remind readers clearly in the main text that the spatial variation revealed by STM images are not a direct

manifestation of the Higgs modes. The authors' scenario seems to have an underlying logic: Higgs mode=amplitude mode=order parameter, and pinning of order parameter can happen. The authors should clarify their logic of how the spatial variation is linked to Higgs modes. This would also be helpful for answering some of the other reviewers questions, e.g., "I have is that the phenomenological model only describes but does not explain the occurrence of the "dual Higgs modes" (reviewer #4)."

2. The authors used phenomenological GL model for theoretical simulation. As presented in the manuscript, the GL theory in the static form, without the need to involve temporal dynamical evolution (amplitude dynamics), already reproduces the spatial variation of CDW amplitude observed in the experiment. The spatial variation emerges owing to the degeneracy of the CDW ground states in this special CuTe material. Given that the static GL equation plus the ground state degeneracy can already explain the results very well, why are the authors still using the more complex scenario involving Higgs modes? The authors should clarify their reasons, and explain to readers whether involving Higgs modes is really a requisite?

In all, the results are interesting and worthy of publication. I am just skeptical about some of the interpretations. For now the authors seems too firm about their proposed scenario, while the theoretical and literature evidence is not solid enough to support their scenario. If the authors' cannot give more convincing evidence (e.g., as suggested by reviewer #4 "Can the occurrence of this phenomenon be derived from the electronic and vibrational structure of CuTe? Or, in other words, can the parameters in the phenomenological model be calculated starting from the atomic structure of the crystal, i.e., from first principles?"), the authors could claim the Higgs scenario as a possible interpretation, rather than a definitive one.

Reviewer #3 (Remarks to the Author):

Reply to the Comments of Reviewer #1

Comment 1) The authors have clarified the doubts and made good changes to the manuscript, which is more clearer from readers.

The explanation for one of the comment is partially satisfactory, for instance, how Higgs modes can unravel the CDW mechanism for other chalcogenides? The author can relook at the comment.

Rest is fine.

(Our reply) We appreciate this comment. Following the advice of the reviewer, we added a short comment on how the present finding of Higgs modes can be helpful for the investigation of the fundamental properties of CDW in chalcogenide materials. In fact, Higgs modes are ubiquitous as the fundamental excitation of a CDW ground state and the present work shows that the Higgs modes can be important to unravel the mechanism of spatial distributions of CDW domains and domain walls, which are widely observed in various chalcogenides (see the following).

“In particular, the interactions between Higgs and phases modes can unravel spatial variations of domains and domain walls, which are ubiquitous in various CDW phases of transition metal chalcogenides.”

Reply to the Comments of Reviewer #2

Comment 1) The experimental results of this manuscript are meaningful and could be of interest of research from CDW and superconductivity communities, but there are still some concerns which I think are not addressed properly in the revised manuscript.

1. I disagree with some of the authors' understanding about Higgs modes. In the first paragraph of Reply to the Comments of Reviewer #2, the authors write: “Amplitude and phase modes of CDW are based on the spatial variation of the CDW amplitude and phase by definition.” Is there any literature supportive of such claim? In my understanding, the definition of Higgs modes in CDW or superconducting systems is the collective excitation of the order parameter, which indeed is dynamical variation along time which in principle cannot be characterized by the static STM probe. According to its definition, Higgs modes have no direct relation to the static spatial distribution. If the Higgs mode could be related to the spatial variation of CDW amplitude, it

should be in an indirect way, which in this work is due to the pinning of CDW amplitude. Besides, the authors also write: “Along the same logic, there should be no fundamental difference between the dynamic amplitude mode and its pinned version.” I think such “logic” is not solid. The two modes are actually different. Phase modes are zero-energy excitations and should be easy for pinning. But Higgs modes have finite energy. In this work, it seems that the degeneracy of two CDW structures are helpful for easier pinning and lead to the intriguing results.

Therefore, if the authors interpret the results in the scenario of Higgs modes, they should remind readers clearly in the main text that the spatial variation revealed by STM images are not a direct manifestation of the Higgs modes. The authors’ scenario seems to have an underlying logic: Higgs mode=amplitude mode=order parameter, and pinning of order parameter can happen. The authors should clarify their logic of how the spatial variation is linked to Higgs modes. This would also be helpful for answering some of the other reviewers questions, e.g., “I have is that the phenomenological model only describes but does not explain the occurrence of the "dual Higgs modes" (reviewer #4).”

Our reply) We appreciate this comment, which we can largely agree with. In the previous reply, we emphasized that the Higgs or phase modes have spatial variations of the order parameters in addition to the time variation and, we interpret, the latter is pinned by defects and the interaction between the Higgs and the phase modes. Note also that the phase modes have excitation gaps in a commensurate CDW. In the present revision, following the advice of the reviewer, we stated explicitly that the present STM image is not a direct manifestation of the Higgs mode in their dynamical form but only probes the spatial variation part as frozen into the soliton lattice.

Comment 2) The authors used phenomenological GL model for theoretical simulation. As presented in the manuscript, the GL theory in the static form, without the need to involve temporal dynamical evolution (amplitude dynamics), already reproduces the spatial variation of CDW amplitude observed in the experiment. The spatial variation emerges owing to the degeneracy of the CDW ground states in this special CuTe material. Given that the static GL equation plus the ground state degeneracy can already explain the results very well, why are the authors still using the more complex scenario involving Higgs modes? The authors should clarify their reasons, and explain to readers whether involving Higgs modes is really a requisite?

Our reply) We appreciate this comment, which we partly agree with. The referee is correct in that the static form of the GL theory is enough to explain the observed STM data (because it observed static patterns) at its face value. Hence, the theory does not need the inclusion of temporal dynamics, if the role of the theory is just to explain the experimental observation. However, it should be noted at the same time that the spatial modulation that we are observing is not the ground state (which will be spatially uniform) and thus it corresponds to the excited modes, which will naturally oscillate in time. This means that, in theory, the involvement of the temporal dynamics is not a mere theoretical complication but a natural consequence of the spatial configuration. In any rate, the referee is correct that in our experiment, we did not observe (at least directly) the time-oscillation of the spatial modulations of amplitudes and the temporal dynamics in theory is not needed to explain the data. Thus, we tone down this part, and change wording accordingly.

In all, the results are interesting and worthy of publication. I am just skeptical about some of the interpretations. For now the authors seems too firm about their proposed scenario, while the theoretical and literature evidence is not solid enough to support their scenario. If the authors' cannot give more convincing evidence (e.g., as suggested by reviewer #4“Can the occurrence of this phenomenon be derived from the electronic and vibrational structure of CuTe? Or, in other words, can the parameters in the phenomenological model be calculated starting from the atomic structure of the crystal, i.e., from first principles?”), the authors could claim the Higgs scenario as a possible interpretation, rather than a definitive one.

Our reply) We agree with this comment and revised our major claim as a possible suggestion since we did not directly measure the dynamics. In particular, we deleted the word “observed” in the title and toned down our claims in abstract, introduction, and summary (see the following).

“Here, we report **the real-space observations, which suggest** two Higgs modes coupled together with a soliton lattice in a solid.”

“**While our observation is limited to the spatial variations of amplitude and phase modes, their connection to dynamical modes is not excluded.**”

“**The dynamics of the present amplitude and phase variations are however not directly observed in the present work and invite further study at an elevated temperature.**”

REVIEWERS' COMMENTS

Reviewer #2 (Remarks to the Author):

The authors have addressed my concerns and made good changes to the manuscript, so I think now it is appropriate for publication.